# LATENT CONSERVATIVE OBJECTIVE MODELS FOR DATA-DRIVEN CRYSTAL STRUCTURE PREDICTION

## ABSTRACT

In computational chemistry, crystal structure prediction (CSP) is an optimization problem that involves discovering the lowest energy stable crystal structure for a given chemical formula. This problem is challenging as it requires discovering globally optimal designs with the lowest energies on complex manifolds. One approach to tackle this problem involves building simulators based on density functional theory (DFT) followed by running search in simulation, but these simulators are painfully slow. In this paper, we study present and study an alternate, data-driven approach to crystal structure prediction: instead of directly searching for the most stable structures in simulation, we train a surrogate model of the crystal formation energy from a database of existing crystal structures, and then optimize this model with respect to the parameters of the crystal structure. This surrogate model is trained to be conservative so as to prevent exploitation of its errors by the optimizer. To handle optimization in the non-Euclidean space of crystal structures, we first utilize a state-of-the-art graph diffusion auto-encoder (CD-VAE) to convert a crystal structure into a vector-based search space and then optimize a conservative surrogate model of the crystal energy, trained on top of this vector representation. We show that our approach, dubbed LCOMs (latent conservative objective models), performs comparably to the best current approaches in terms of success rate of structure prediction, while also drastically reducing computational cost.

## 1 INTRODUCTION

Data-driven optimization problems arise in many areas of science and engineering. In these settings, we have an unknown function that we would like to optimize with respect to its inputs, provided only with a dataset of input-output pairs. Examples include drug design, where inputs might be molecules and outputs are the efficacy of a drug, protein design, where inputs correspond to protein sequences and outputs are some metric such as fluorescence (Sarkisyan et al., 2016) or, as in our experiments, prediction of crystal structures, where inputs consist of crystal structures and outputs correspond to their formation energy. Such data driven optimization problems present several challenges. First, naïvely training a predictive model to predict the output from the input and then optimizing against such a model may lead to exploitation: a sufficiently strong optimizer can typically discover inputs that lead any learned model to extrapolate erroneously, and then exploit these errors to find inputs that "fool" the model into making the desired predictions. Second, even if a model can be suitably robustified, many of the most important design and optimization problems in science and engineering, including crystal structure prediction, require optimizing over complex sets and non-Euclidean manifolds, such that naïvely applying gradient-based methods in the input space is unlikely to result in a meaningful improvement.

In this paper, we study these challenges in the context of crystal structure prediction (CSP) Woodley and Catlow (2008). Crystals are a class of solid-state materials characterized by the periodic placement of atoms. These structures form the basis of a wide variety of applications such as designing super-conductors, batteries Yamashita et al. (2016), and solar cells Walsh et al. (2012). Computationally identifying stable crystal geometries given a particular chemical formula typically involves minimizing (an estimate of) the crystal's formation energy to find the minimal energy structure. Conventional approaches to this problem rely on slow and compute-intensive DFT simulators Chermette (1998), but more recent machine learning approaches dispense with DFT-based simulators and use databases of structures and their corresponding energies to train models that estimate crystal formation energy directly Gasteiger et al. (2021); Klicpera et al. (2020). However, the CSP problem suffers from both issues outlined above: crystal structures typically exhibit highly

complex geometries characterized by periodicity of the lattice that forms the crystal and discrete (e.g., number and types of atoms in the chemical compound) and continuous features (e.g., positions of atoms in 3D space), which make it hard to produce reliable estimates of energies across the entire manifold of possible structures. Optimizing the structure using such inaccurate models then bears the risk of the optimization procedure "exploiting" these inaccuracies, resulting in structures that erroneously appear promising in this learned model but are not stable.

In this paper, we aim to develop a data-driven optimization approach to overcome these challenges. First, to avoid the complexities associated with optimization over the complex manifold of crystal structures that consists of both discrete and continuous objects, our optimization procedure utilizes a crystal diffusion variational auto-encoder (CD-VAE) (Xie et al., 2021) to convert crystal structures into latent representations, which are much more amenable to simple gradient-based optimization. Second, to prevent the optimizer from getting "fooled" by the errors in the learned surrogate model, we extend conservative objective models (Trabucco et al., 2021b), a robustification procedure, to our surrogate energy prediction model. This procedure explicitly pushes down over-estimated out-of-distribution designs in the latent space. Using a combination of these techniques, we develop a method for finding stable crystal structures that alleviates the time and compute costs associated with using DFT simulators, while also addressing the inaccuracies in a purely offline approach for designing crystal structures.

Our main contribution is a data-driven optimization approach, that we call latent conservative objective models (LCOM), for the problem of crystal structure prediction for solid materials. Our method leverages both advances in generative modeling over periodic solid-state materials for latent space learning (Xie et al., 2021) and recent advances in model-based optimization for robustifying the learned model to make it amenable to direct optimization of formation energies (Trabucco et al., 2021b). We summarize the approach in Figure 1. We instantiate our approach, latent conservative objective models (LCOMs), using crystal diffusion variational auto-encoders (CD-VAE) Xie et al. (2021) for learning the latent space and conservative objective models (COMs) Trabucco et al. (2021b) for optimization. Empirically, we demonstrate that LCOMs are able to match the performance of the best prior method from Cheng et al. (2022) while significantly reducing the total wall-clock time needed for optimization by 40 times. In particular, a single optimization cycle in our framework takes an average of 2 seconds. This allows our model to provide predictions for more than 100 compounds in 4 minutes, much faster than prior works.

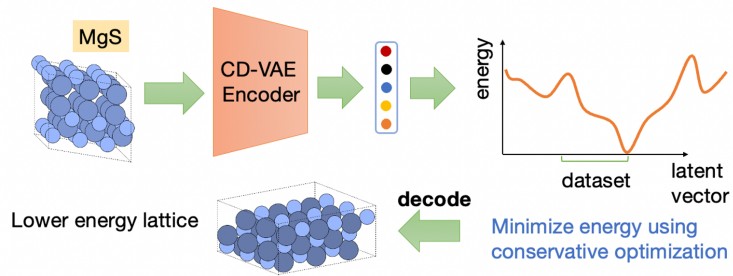

Figure 1: **Overview of LCOMs.** We train a graph-based CD-VAE to construct a latent space the represents crystal structure, conditioned on the molecular structure of the compound. Different points in this latent space correspond to different crystal structures, and we then optimize over the structure with simple gradient-based optimization methods operating on this latent space. To do so, we train a surrogate energy prediction model on the learned latent space via conservative training (Trabucco et al., 2021b) to make it robust on out-of-distribution inputs, thus preventing the optimizer from discovering latent space points far from the training data for which the energy predictions yield erroneously low energies. The optimized latent vector is then decoded into a structure. Since the entire optimization is performed in the latent space, the comparatively complex encoder and decoder only need to be used once during optimization (to encode the initial structure and decode the final one).

## 2 BACKGROUND ON CRYSTAL STRUCTURES AND MATERIALS

In this section, we present the background definitions associated with crystals and solid-state materials. A crystal is a solid-state material characterized by a periodic placement of its constituents, which are chemical elements. The stoichiometry or the composition of a crystal, like NaCl, consists of the elements that make up the solid-state material (i.e., Na and Cl in this case) and in what ratio. In real-world applications of solid-state materials it is not enough to develop a material with a suitable

chemical composition, but we must also account for the crystalline periodic structure of the solid and the atoms' positions with respect to it to assess the stability of a given compound.

Mathematically, we can describe the periodic structure of a chemical by defining its lattice $L$ in 3D space, which repeats periodically. To characterize a lattice, we define its base vectors $\mathbf{v}, \mathbf{w}, \mathbf{z}$. Every point in the lattice is a linear combination of these vectors using only integer coefficients.

$$p \in L \iff \exists n, m, k \in \mathbb{Z} \,|\, p = n\mathbf{v} + m\mathbf{w} + k\mathbf{z}. \tag{1}$$

Given a lattice $L$, the unit cell is the volume of 3D space contained between the base vectors, defined formally as follows:

$$\left\{ p \in \mathbb{R}^3 \,|\, \exists x, y, z \in [0, 1], \, p = x\mathbf{v} + y\mathbf{w} + z\mathbf{z} \right\}. \tag{2}$$

We can obtain the entire lattice of a crystal by periodically repeating this unit cell in 3D space. Given a lattice in 3D space, a crystal is additionally characterized by how many atoms $n$ are in the unit cell. We observe that the number $n$ is always a multiple of the number of elements in the chemical composition of the material. For example, for a formula $MgO_3$, which consists of 4 atoms, we can have 4, 8, 12, or generally $4k$ elements in a unit cell (one element corresponding to one atom), but not 3 or 6 elements, which are not a multiple of 4.

Finally, we can describe atoms' types and positions with two matrices $A \in \mathbb{R}^{n \times 128}$, $X \in \mathbb{R}^{n \times 3}$. The matrix $A$ identifies different atoms in the unit cell using a one-hot representation. Specifically, $A_i$ is a vector with a 1 at position $Z$ corresponding to the atomic number of the $i$-th element, and 0 everywhere else. The matrix $X$ provides fractional coordinates for the atoms. These are coordinates between 0 and 1 with respect to the basis defined by the lattice base vectors. More specifically, the vector $X_i$ tells us that the $i$-th element is at $X_i^1 \mathbf{v} + X_i^2 \mathbf{w} + X_i^3 \mathbf{z}$ in the unit cell.

To summarize, a crystal is defined by three quantities: **(1)** A $3 \times 3$ matrix $L$ representing the lattice, the rows of which corresponding to the base vectors of the lattice; **(2)** number ($n$) and types ($A \in \mathbb{R}^{n \times 128}$) in the chemical; and **(3)** atoms positions $X \in \mathbb{R}^{n \times 3}$, which is specified in terms of fractional coordinates between 0 and 1. In the following sections, we will denote a crystal with the variable $\mathbf{x}$ and we will use subscripts to refer to lattice parameters $\mathbf{x}_L$, atoms' types $\mathbf{x}_A$, and fractional coordinates $\mathbf{x}_X$. We finally remark that the majority of crystal and lattice configurations for a given chemical compound are "unstable" and would collapse to a different configuration when synthesized.

## 3 PROBLEM STATEMENT, DATASET, AND EVALUATION

Crystal structure prediction (CSP) is the problem of finding a crystal of a given chemical composition (e.g. NaCl, or $MgO_3$) that attains the global minimum of the crystal formation energy. Such a crystal is sythesizable and can be utilized for various downstream applications.

**Problem 3.1 (Crystal structure prediction).** Given a chemical composition $c$, find the crystal $\mathbf{x}^* = (L^*, A^*, X^*)$ with lattice matrix $L^*$, atom types $A^*$, and atom coordinates $X^*$ such that $\mathbf{x}^*$ minimizes the formation energy function $E$ for the chemical composition.

$$\mathbf{x}^* = \operatorname{argmin}_{\mathbf{x}} E\left(\mathbf{x}, c\right).$$

**Why is solving CSPs hard?** Only very few crystal structures are actually stable and only one of these stable structures is at a global minimum, which makes crystal structure prediction equivalent to searching for a "needle in a haystack". The difficulty of solving a CSP is further compounded by the fact that the search space over all possible crystal structures for a given chemical composition is quite complex and non-Euclidean. This is because there is no one-to-one correspondence between the matrices $(A, X)$ and lattices $L$. Given a lattice matrix $L$, every other matrix that is rotationally equivalent or permutation equivalent represents the same lattice. Moreover, reducing the design space from all possible structures to only stable ones changes the manifold of designs considerably.

In principle, we could always attempt to find the globally optimal crystal structure by evaluating many possible candidate structures against a simulator, but simulators for CSP are typically based on density functional theory (DFT), and generally these are extremely slow in terms of the wall-clock time. Therefore, we intend to solve this problem using only existing static datasets (OQMD (Saal et al., 2013) and MatBench Dunn et al. (2020)). that contain several (sub-optimal) crystal structures for a variety of chemical compounds along with their corresponding formation energies. With no access to the simulator, our goal is to find a globally optimal crystal structure given a new chemical compound. We describe our procedure for constructing the dataset and our evaluation protocol next.

**Datasets for training.** To robustly evaluate our method, we consider two training scenarios with different datasets: the OQMD dataset (Saal et al., 2013) and the MatBench dataset Dunn et al. (2020), both of which consist of the crystal structures and formation energies for obtained via DFT simulations. Every sample in the dataset represents a stable crystal structure $\mathbf{x}$ (i.e., a crystal at a local minimum of energy) computed via numerical relaxation (Hafner, 2008), together with its chemical composition $c$ and formation energy $E(\mathbf{x}, c)$. We chose these datasets because of their large size (OQMD has more than 1 million examples), and the availability of more than one stable structure per chemical, all of which are not at their global optimum.

**Held-Out evaluation datasets.** We evaluate our offline optimization approach in terms of its efficacy in discovering the globally optimal structure for a given chemical compound. To this end, following the protocol of Cheng et al. (2022), we construct a held-out test set consisting of some compounds and the associated globally optimal crystal structures (Table 1) and utilize this set for evaluations.

**Evaluation protocol.** Following the evaluation protocol in prior works Xie et al. (2021); Cheng et al. (2022), we test our method in terms of its efficacy in recovering the globally optimal structure on 26 of the 29 compounds in (Cheng et al., 2022), where the 3 remaining compounds are omitted because they cannot be simulated with GPAW to compute their ground truth energy values. These compounds are listed in Table 1. For each of these compounds, we run our optimization process to convergence and check the final energy of the optimized design. We compare the optimized energy against the known global minimum energy. We consider it a success if the final energy of the optimized structure produced by our approach recovers the value of the known global minimum, up to a predefined noise threshold of $20\%$ of the ground truth minimum energy.

To stress-test our gradient-based approach, we seed the optimizer with a random stable initial crystal structure. We compute this initial stable structure by running simulations in the GPAW (Enkovaara et al., 2011) simulator, an open-source DFT simulator fully integrated into python packages for chemistry like ase and pymatgen. Concretely, we utilized the following protocol for obtaining this initial crystal structure: **(i)** for every compound, we first select the number of atoms corresponding to the optimal compound design as listed in the materials project database, **(ii)** we then randomly initialize the lattice matrix and the atom coordinates, and **(iii)** we then run the process of structure relaxation in the simulator to obtain the closest local minimum (i.e., a closeby structure that is stable).

## 4 LCOMs: Latent Conservative Objective Models

To design crystal structures with the lowest possible energy entirely from an existing dataset of only sub-optimal structures, we utilize techniques from offline model-based optimization. Directly applying these techniques (Trabucco et al., 2021b; Yu et al., 2021; Qi et al., 2022) for optimizing crystals is non-trivial as these methods typically employ optimization procedures that iteratively make local changes to the design (e.g., gradient-based updates or random mutations) to optimize a "surrogate" estimate of the objective function. Such iterative procedures fall short when optimizing over non-smooth geometries and non-Euclidean manifolds like that of crystals. To alleviate this issue, we propose an approach for offline optimization that first learns a latent vector representation for a crystal structure, then performs data-driven optimization in this vector space, and finally maps back the resulting outcome to a valid crystal structure. We outline each part of this procedure below.

### 4.1 Transforming Crystal Structures to a Latent Representation

Which sort of a latent representation space is especially desirable for CSP? Since one of the central challenges in our problem is the abundance of invalid or infeasible structures in the space of all possible crystals, it is very desirable to learn latent representations that only encode valid and feasible structures. Once we learn such a space, we can directly perform offline optimization in this latent space. If we can ensure that every possible latent vector corresponds to some feasible crystal structure, then we are guaranteed to at least prune out the possibility of finding infeasible structures during the optimization process. To this end, we train a crystal diffusion variational auto-encoder (CD-VAE) on our training dataset for various chemical compositions, and then run offline optimization in the latent space of this auto-encoder. Since our training dataset only consists of stable structures, the decoder of a well-trained CD-VAE should map latent vectors to the manifold of stable crystal structures only, which would greatly benefit optimization by enabling the optimizer to move in a much smaller manifold. Below we describe the architecture and the training objective for the CD-VAE.

**CD-VAE.** A CD-VAE (Xie et al., 2021) is composed of three parts: a graph neural network (GNN) encoder $\text{PGNN}_{\text{ENC}}$ that takes a crystal $\mathbf{x}$ as input and outputs a latent vector representation, an NN predictor $\text{MLP}_{\text{AGG}}$ that outputs lattice parameters $\mathbf{x}_L$ from its encoded representation $\text{PGNN}_{\text{ENC}}(\mathbf{x}, c)$,

and a GNN diffusion denoiser $PGNN_{DEC}$ that takes a random noisy crystal $\tilde{x}$ and a latent encoding $PGNN_{ENC}(x, c)$ as inputs, and outputs forces to apply on the atoms coordinates $\tilde{x}_X$ to build the original crystal $x$ via a diffusion process. Following Xie et al. (2021), the encoder $PGNN_{ENC}$ uses a DimeNet++ Klicpera et al. (2020) architecture. Likewise, the decoder $PGNN_{DEC}$ uses a GemNet-dQ Gasteiger et al. (2021) architecture. During decoding, CD-VAE initializes a structure with random lattice and coordinates and utilizes Langevin dynamics Song and Ermon (2019) to gradually recover the stable structure represented by the latent vector.

To make the notation compact, we will refer to the $PGNN_{ENC}$ as $\phi$, and thus the latent representation of a crystal $x$ with chemical composition $c$ will be denoted by $\phi(x, c)$. We will use the notation $PGNN_{DEC}(z)$ to denote the structure obtained after applying the denoising process with latent vector $z$. Akin to a variational auto-encoder (Kingma and Welling, 2013), CD-VAE (Xie et al., 2021) is also trained to maximize the likelihood of crystal structures seen in the dataset, agnostic of the energy objective that we wish to optimize.

**Training objective for the latent representation.** We follow the training objective utilized by the CD-VAE (Xie et al., 2021). The first term in this objective is the reconstruction error over lattice parameters, formally defined as:

$$\mathcal{L}_{AGG}\left(MLP_{AGG}\left(\phi(x, c)\right), x_L\right) = \|x_L - \phi(x, c)\|^2. \tag{3}$$

Akin to a VAE (Kingma and Welling, 2013), we also include a loss term minimizes the KL-divergence between a normal distribution over the latent representation induced by the encoder (with mean $\phi(x, c)$ and a learned standard deviation) and a standard multi-dimensional normal distribution.

The decoder of the CD-VAE is a denoising diffusion model (Ho et al., 2020) that attempts to transform an input latent vector into the corresponding crystal structure, starting from a random structure $\tilde{x}$, which is iteratively refined via the diffusion process. Succinctly, the objective for training this term is

$$\mathcal{L}_{DEC}\left(\tilde{x}, x, \phi(x, c)\right) = \frac{1}{2L}\sum_{j=1}^{L}\left[\mathbb{E}_{\sigma_j}\left\|PGNN_{DEC}\left(\tilde{x}, \phi(x, c)\right) - \frac{d\left(x_X, \tilde{x}_X\right)}{\sigma_j}\right\|\right], \tag{4}$$

where $\{\sigma_j\}_{j=1}^{L}$ are noise schedule scalars for the diffusion process and are in a geometric sequence with common ratio greater than 1. Finally, we remark that we do not utilize terms for reconstructing types or the number of atoms (i.e., $x_A$ or $x_n$) because our optimization procedure only aims to optimize over other parameters of the crystal lattice so the number and types of atoms are fixed.

## 4.2 CONSERVATIVE OPTIMIZATION IN LATENT SPACE

Once the encoder of the CD-VAE is trained, we can then train a surrogate model, $\widehat{E}_\theta(\phi(x, c), c)$ to estimate the formation energy $E$ of a crystal structure $x$ for a given chemical composition, $c$ via standard supervised regression. Then, we can simply optimize the crystal structure to maximize the outputs of this surrogate model. However, as prior works (Kumar et al., 2020; Trabucco et al., 2021b) note, this simple strategy often fails at finding optimized designs due to the exploitation of errors in the learned surrogate model by the optimizer. To address this issue, we extend the conservative objective models (COMs) technique for optimizing crystals in the learned latent space.

**Training latent space conservative models.** To prevent the optimization procedure from exploiting inaccuracies in this learned surrogate model, we apply an additional regularizer from Trabucco et al. (2021b); Kumar et al. (2021) to robustify the surrogate model. This regularizer mines for adversarial vectors in the latent space $z^+$ that appear to have very low energies $\widehat{E}_\theta(z^+, c)$ under the learned surrogate model, and then explicitly pushes up the predicted energy $\widehat{E}_\theta(z^+, c)$ on such adversarial $z^+$. Following the COMs approach (Trabucco et al., 2021b), we interleave the training of the learned surrogate model $\widehat{E}_\theta$ with an optimization procedure $Opt(\widehat{E}_\theta, c)$ that seeks to find the aforementioned adversarial vectors $z^+$ that optimize the current snapshot for the surrogate model, for a given chemical composition $c$. After these adversarial vectors are found, the training procedure explicitly pushes up the energy output of the surrogate model on such points. To compensate for the effect of increasing the learned energy values in an unbounded manner on all latent vectors, we additionally balance the push up term by pushing down the energy values on the latent representations induced by crystal structures in the data. This idea can be formalized into the following loss for training $\widehat{E}_\theta$:

$$\min_\theta \ \mathbb{E}_{c,x\sim\mathcal{D}}\left[\left(\widehat{E}_\theta(\phi(x, c), c) - E(x, c)\right)^2\right] - \alpha\left(\mathbb{E}_{c,x\sim\mathcal{D}}\left[\mathbb{E}_{z^+\sim Opt(\widehat{E}_\theta, c)}[\widehat{E}_\theta(z^+, c)] - \widehat{E}_\theta(\phi(x, c), c)\right]\right).$$

We will discuss the formulation for Opt below. Crucially, unlike COMs (Trabucco et al., 2021b), which directly runs gradient descent in the input space, our approach operates in the latent space.

**Optimizing in the latent space.** Once a conservative surrogate model $\widehat{E}_\theta(z, c)$ is obtained using the above training procedure, we must now optimize this model to obtain the best possible structures. The optimization procedure Opt that was used to obtain adversarial latent vectors in the training objective above can then be repurposed to obtain optimized latent vectors once the latent conservative model is trained. Since the latent space $z$ is a continuous Euclidean vector space, for any given chemical composition $c$, our choice of Opt is to run $T$ rounds of gradient descent on the surrogate energy $\widehat{E}_\theta(z, c)$ with respect to the latent vector $z$, starting from the latent vector $z_0$ corresponding to a random initial crystal structure. For a given $c$, this procedure can be formalized as follows:

$$z_{k+1} \leftarrow z_k - \alpha \nabla_z \widehat{E}_\theta(z, c), \tag{5}$$
$$\text{where} \quad z_0 \sim \phi(\mathbf{x}_0, c), \ \ \mathbf{x}_0 \sim \mathcal{D}.$$

Once this optimization procedure is run for $T$ steps, we pass the final latent vector $z_T$ to the decoder of the pre-trained CD-VAE to obtain the optimized crystal structure: $\widehat{\mathbf{x}}^* = \text{PGNN}_{\text{DEC}}(z_T)$.

### 4.3 IMPLEMENTATION DETAILS

For obtaining the latent space, we train a CD-VAE identically to Xie et al. (2021) on our datasets, following their implementation details for the encoder and the decoder. After training the CD-VAE, we encode molecule structures from the dataset into vectors, and these vectors are then used as inputs for training the optimization model. For training LCOMs, we represent the conservative objective model $\widehat{E}_\theta(\phi(\mathbf{x}, c), c)$ as a neural network with two hidden layers of size 2048 each and leaky ReLU activations. For computing $z^+$, we perform one gradient descent step on the vector $z$ from input latent space. We perform 50 gradient steps and get an optimized vector in the latent space. We then decode these latent vectors into an optimized crystal structure. For reporting statistically robust results, we repeat the optimization procedure for three seeds and average over the resulting energy value.

## 5 RELATED WORK

One widely studied optimization-based approach to CSP utilizes evolutionary algorithms Lonie and Zurek (2011); Oganov et al. (2011). For instance, USPEX Glass et al. (2006) is an algorithm that uses a variety of heuristic strategies for iteratively evolving structures directly in the space of the crystal parameters. Each of these intermediate structures need to be evaluated against the simulator, which repeatedly involves running relaxation to the nearest stable structure. This extensive use of simulation makes such an approach computationally impractical, necessitating offline learning-based approaches like our method that do not require any simulation.

Due to the availability of large public datasets such as the materials project database and the open catalyst project (Chanussot et al., 2021), recent works develop learning-based approaches for solving CSPs. Another subset includes methods that use different types of evolutionary optimizers to optimize a GNN-based surrogate energy model instead of the ground-truth energy function. This includes methods based on random search (Cheng et al., 2022), particle swarm optimization (Clerc, 2010), and Bayesian optimization (Pelikan et al., 1999). In contrast, our method prescribes the use a conservative surrogate model of the energy function, that takes as input a latent representation of the crystal. As we show in our experimental results, both of these aspects are crucial for CSP.

Another line of prior work learns generative models for graph data including, but not limited to crystal structures. This includes methods that leverage variational auto-encoders (Simonovsky and Komodakis, 2018) normalizing flows (Satorras et al., 2021), generative adversarial networks (Kim et al., 2020), recurrent neural networks Grisoni et al. (2020), reinforcement learning techniques (Pereira et al., 2020) or a combination of auto-encoders and diffusion models, for example, the CD-VAE (Xie et al., 2021), that we build upon. While these approaches aim to model the manifold the graph-structured data and our approach utilizes the latent space learned by one such approach, CD-VAE (Xie et al., 2021), our goal of optimizing the structure is distinct from the goal of modelling the data.

Model-based optimization (MBO) refers to the problem of optimizing an unknown function by constructing a surrogate model. Bayesian optimization represents one of the most widely known classes of MBO methods (Snoek et al., 2015; 2012; Ghavamzadeh et al., 2015), but classically MBO requires iteratively sampling new function values, which can be very expensive when evaluating a crystal structure's energy requires an expensive simulation process. More recently, offline MBO methods, sometimes referred to as data-driven optimization, have been proposed to optimize designs

| | OQMD | | | | Baselines | | MatBench | | | |
|---|---|---|---|---|---|---|---|---|---|---|
| Compounds | RAS* | PSO* | BO* | **LCOMs** | SL | CD-VAE | RAS* | PSO | BO | **LCOMs** |
| LiF | | ✓ | ✓ | ✓ | | | ✓ | ✓ | | ✓ |
| NaF | ✓ | ✓ | ✓ | | ✓ | | | | | |
| KF | ✓ | | ✓ | ✓ | | ✓ | | | | ✓ |
| RbF | | | | ✓ | ✓ | ✓ | ✓ | | | ✓ |
| CsF | | | | ✓ | | ✓ | ✓ | | ✓ | ✓ |
| LiCl | | | | | | | | ✓ | | |
| NaCl | ✓ | ✓ | ✓ | ✓ | ✓ | ✓ | ✓ | ✓ | ✓ | ✓ |
| KCl | ✓ | | | ✓ | | | ✓ | | ✓ | ✓ |
| RbCl | ✓ | | | ✓ | | | ✓ | ✓ | | ✓ |
| CsCl | ✓ | | ✓ | ✓ | | | ✓ | | | ✓ |
| BeO | | ✓ | ✓ | ✓ | | ✓ | | ✓ | ✓ | ✓ |
| MgO | ✓ | | ✓ | | | | ✓ | | ✓ | ✓ |
| CaO | ✓ | | ✓ | ✓ | | | ✓ | | | ✓ |
| SrO | ✓ | | ✓ | ✓ | | | ✓ | ✓ | | ✓ |
| BaO | ✓ | | ✓ | | | | ✓ | | | ✓ |
| ZnO | | ✓ | ✓ | ✓ | | | | ✓ | ✓ | ✓ |
| CdO | | | | | | ✓ | ✓ | | | |
| BeS | ✓ | ✓ | ✓ | ✓ | | | ✓ | ✓ | | ✓ |
| MgS | ✓ | | | | ✓ | | ✓ | ✓ | | ✓ |
| CaS | ✓ | | ✓ | ✓ | | | ✓ | ✓ | ✓ | ✓ |
| SrS | ✓ | | | ✓ | ✓ | | ✓ | ✓ | ✓ | ✓ |
| BaS | ✓ | | ✓ | | | | ✓ | ✓ | ✓ | |
| ZnS | ✓ | | ✓ | ✓ | | | | ✓ | ✓ | ✓ |
| CdS | | | ✓ | | | | | ✓ | ✓ | |
| C | ✓ | | | | | | ✓ | | | |
| Si | | | | | | | | | | |
| Accuracy | 17/26 | 6/26 | 16/26 | 16/26 | 5/26 | 6/26 | 18/26 | 13/26 | 10/26 | **19/26** |

Table 1: **Evaluation of LCOMs and other prior crystal structure prediction methods** in terms of the accuracy of discovering the globally optimal structure for 26 compounds. Check marks indicate successful discovery as per our criterion. Note crucially that while we utilize a threshold on optimization energy to determine success, baseline methods marked with * in this table follow prior works and utilize a manual inspection protocol as discussed in the footnote in Section 6.

based entirely on previously collected *static* datasets (Brookes et al., 2019; Trabucco et al., 2021a; Kumar and Levine, 2019; Trabucco et al., 2021b; Qi et al., 2022). Our work builds on these methods, and is most closely related to the COMs algorithm proposed by Trabucco et al. (2021b). However, while prior offline MBO methods focus on robustifying the surrogate model in the design space directly, we integrate these approaches with latent space optimization that makes it possible to optimize over the manifold of only stable structures, while still using a simple gradient optimizer.

## 6 EXPERIMENTAL EVALUATION

The goal of our experimental evaluation is to evaluate the efficacy of LCOMs for crystal structure prediction by answering the following questions: **(1)** Can LCOMs successfully optimize in the latent representation space?, **(2)** Do LCOMs manage to effectively recover the optimal energy structure up to a pre-defined threshold of accuracy?, and **(3)** Does LCOM drastically reduce simulation wall-clock time compared to prior methods? To answer these questions, we evaluate LCOMs against prior methods following the protocol from Section 3 and then perform some diagnostic experimental studies that we will discuss in this section.

**Comparing LCOMs with baselines and prior methods**. We compare LCOMs to two methods from prior work (Cheng et al., 2022): particle-swarm optimization (PSO), and Bayesian optimization (BO). We also study two baseline methods: a method that does not run any optimization in the latent space and simply constructs a stable structure for a given chemical compound via the decoder ("CD-VAE"), and a method where the crystal is optimized with a naïve supervised learning model in the latent space of the CD-VAE via gradient descent ("Supervised learning; SL"). Note that the latter is similar to LCOMs, but the surrogate energy prediction model is not trained with any conservatism, but rather with only standard supervised regression. We evaluate these methods on the 26 chemical compounds in our evaluation dataset, and present the results in Table 1. During evaluation, we mark a crystal structure successful if the energy of the optimized structure is close to the energy of the best known globally optimal structure up to a certain threshold. This threshold is defined as an upper bound of 0.2 on the quantity $(E(\mathbf{x}, c^*) - E(\mathbf{x}, \hat{c})) / |E(\mathbf{x}, c^*)|$ to account for imprecision in the simulator, where $c^*$ is the ground truth optimal crystal and $\hat{c}$ is the optimized crystal discovered by the optimizer.

The results in Table 1 show that when trained on OQMD, our method improves significantly over naïvely optimizing in the CD-VAE latent space without conservative training (supervised learning; SL), and also that it is competitive with the prior state-of-the-art methods RAS, PSO, and BO, exceeding the performance of the PSO baseline and matching BO and RAS, without needing any simulations (we will quantify the benefits on wall-clock time soon). A similar trend also holds for the MatBench dataset in Table 1, indicating that LCOMs is performant for different choices of the training data. On MatBench, LCOMs outperforms both PSO and BO methods by a large margin. We also note that BO and RAS perform comparably to LCOMs when trained on the OQMD dataset, when results for these prior comparisons are reported using the evaluation metric in (Cheng et al., 2022)[1]. Due to the difference in evaluation criteria, these comparisons to LCOMs are not as exact like in MatBench, where both BO and PSO are evaluated using the same criteria as LCOMs.

**Does LCOM improve over prior methods in terms of wall-clock time?** Our method optimizes the crystal structure in the latent space of the CD-VAE, using gradient-based optimization. One advantage of this approach is computational efficiency, since the complex graph-based component of the pipeline is only used during the encoding and decoding stages at the beginning and end of the optimization, rather than at each optimization step. Hence, we measure the wall-clock time needed to run optimization with LCOMs comparatively against other prior methods in Table 2. Observe that while utilizing a graph neural network (GNN-BO) reduces the wall-clock time needed by about $875\times$ compared to DFT-PSO that queries the simulator for every design, LCOMs further reduces the wall-clock time $40\times$ by running optimization in the latent space of a CD-VAE, which does not require running expensive message passing loops of a graph neural network encoder but rather runs relatively faster forward passes through small MLPs. This indicates that LCOMs not only discovers are more optimal crystal structure, but it does so $40\times$ faster than the best prior method.

|  | DFT-PSO | GN-BO | **LCOMs** |
|---|---|---|---|
| Optimization time per structure (seconds) | 70000 | 80 | 2 |

Table 2: **Comparing wall-clock time for different methods.** Observe that not using a simulator reduces the wall-clock time from 70000 seconds to 80 seconds per structure, and further utilizing a latent space surrogate model in LCOMs instead of a graph neural network model cuts down the total time further by $40\times$ to 2 seconds.

**How does conservative training influence optimization?** In the next set of experiments, we aim to understand how conservative training of the surrogate model influences the behavior of gradient descent optimization in the CD-VAE latent space. If the energy model is trained naïvely (i.e., with standard supervised regression), it will make arbitrarily erroneous predictions when queried on adversarial, out-of-distribution crystals that differ too much from the training data. Some of these errors will be under-estimation errors, and therefore, a strong optimizer would be able to exploit these errors to find points in the latent space for which the model erroneously predicts arbitrarily low energies. Empirically, we evaluate the performance of optimized crystals obtained by running 50 gradient steps of optimization on the learned surrogate models starting from an initial structure. Specifically, we compute the relative improvement in energy values after optimization, formally calculated as $(E(\mathbf{x}, c_0) - E(\mathbf{x}, \hat{c}))/|E(\mathbf{x}, c_0)|$, where $c_0$ denotes the initialization and $\hat{c}$ denotes the optimized structure in Figure 3.

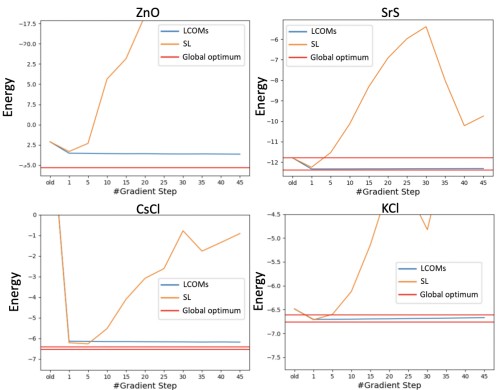

Figure 2: **Comparing the energy of intermediate structures observed over the course of optimization** with LCOMs (blue) and non-conservative models (orange). Note that while the non-conservative model gets exploited as more steps of gradient-based optimization are performed, structures discovered by conservative LCOMs attain lower energies after gradient descent, and the final structures are close to the global optimum (marked as red in the plot above).

---

[1]Our evaluation protocol for determining the optimality of a structure uses an energy threshold. This is a contrast to Cheng et al. (2022), which adopts a manual inspection approach to check for equality between optimized and optimal structures as their evaluation criterion. As such, comparisons between our work and that of Cheng et al. (2022) marked with an asterisk (*) should be made with an understanding of this fundamental difference in evaluation methodology.

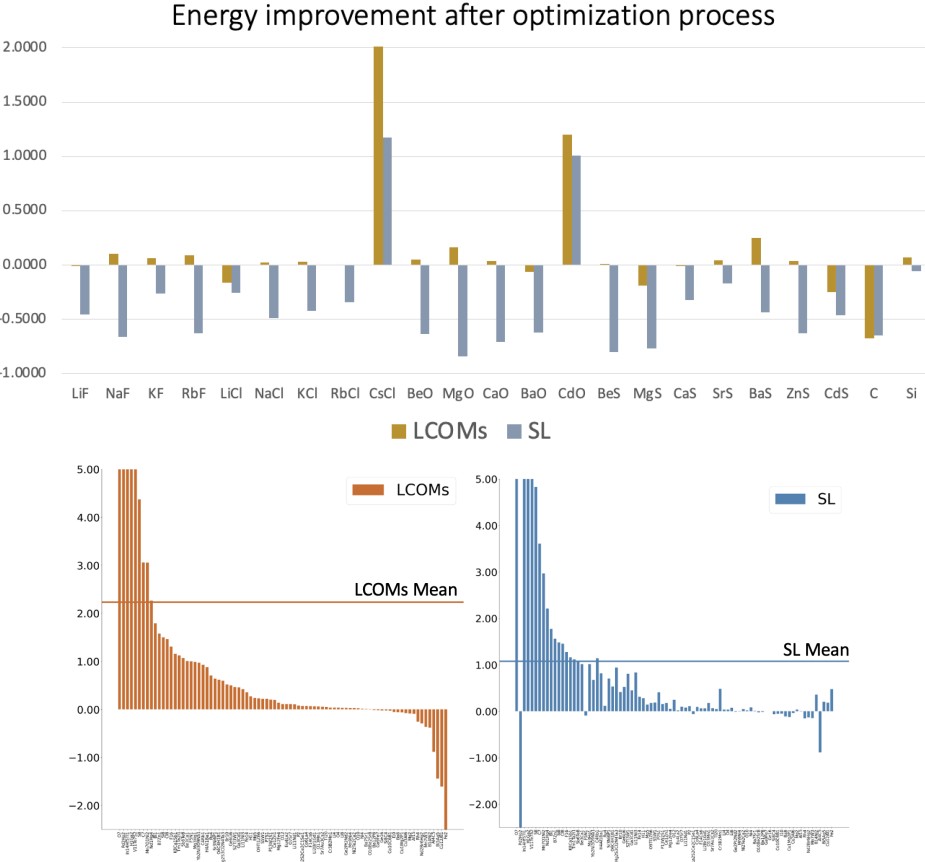

Figure 3: **Comparison of energy improvements produced by LCOMs and the non-conservative supervised learning (SL) baseline. Top:** Energy improvement for 25 compositions, when training on the OQMD dataset; **Bottom:** energy improvement over a set of 83 chemical compositions, when training on the MatBench dataset. Note that structures found by LCOMs achieve better formation energy after optimization, while the non-conservative supervised learning model actually leads to structures with worse energy values (negative improvement). The quantity on the y-axis represents the percentage of improvement (reduction) in the energy value. These results indicate that conservative training is essential for successfully instantiating a method with gradient-based latent space optimization for crystal structure prediction.

Observe that while LCOMs generally produces positive improvement, the non-conservative model leads to *negative* improvement: the optimized structures are generally worse than the random structure at initialization. This indicates that conservative training is critical for latent space optimization to work. We also perform a more fine-grained analysis, where we plot the trajectory of evolution of the energy values over each round of optimization in Figure 2. Observe that for the non-conservative model (orange), the energy increases over the course of optimization indicating that the optimizer is exploiting errors in the learned model. This exploitation is absent for LCOMs (in blue), indicating that conservatism is crucial for attaining good performance.

## 7 DISCUSSION AND FUTURE DIRECTIONS

We presented a method for offline optimization that uses the latent space of a CD-VAE to perform smooth gradient-based optimization of complex structures, with application to crystal structure prediction. Our method combines concepts from conservative objective models that robustify predictive models to make them amenable to gradient-based optimization, with a latent-space generative models of graphs, enabling us to use simple gradient-based optimization methods. Experiments show that our method can successfully optimize the formation energy and recover the optimal structure of a chemical compound with a good level of accuracy, comparing favorably with existing approaches, while tremendously reducing computation time. An interesting avenue for future work is to study the efficacy of LCOMs in more problems in computational chemistry. Another direction for future work is to use the best performing models to predict the optimized structure for novel chemicals and validate the predictions experimentally.

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

# Appendices

## A    ADDITIONAL ABLATION STUDY

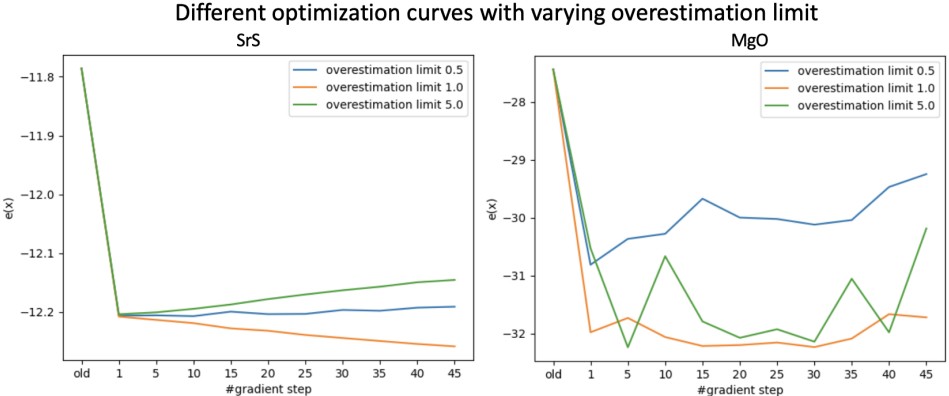

Figure 4: **Ablation study.** The overestimation threshold, $\tau$, is the factor controlling the level of conservatism imposed by LCOMs. The above plot shows the performance of the crystal structures found by LCOMs by varying this threshold for two sample compounds: SrS and MgO.

Since our method builds on existing conservative optimization algorithms, one of the main hyper parameters of our method is the coefficient $\alpha$ controlling the strength of the conservatism regularizer. In the practical instantiation of COMs (Trabucco et al., 2021b), this hyperparameter is replaced by its Lagrangian dual Equation 6 in Trabucco et al. (2021b)), and the corresponding hyperparameter in the practical algorithm is $\tau$, the threshold of allowed over-estimation on adversarial examples (in our case, adversarial latent vectors). A smaller $\tau$ enforces a stricter upper bound on the allowed amount of distribution shift, whereas a larger $\tau$ does not penalize distributional shift. As a result, energies of produced designs would be close to the energy in the dataset when the coefficient $\tau$ is small, but also get exploited when $\tau$ is too large. An intermediate value of $\tau$ is expected to likely lead to the most favorable results.

As shown in Figure 4, an intermediate value of $\tau$ (e.g., 1.0 in this case) leads to the best results as more gradient steps are performed to optimize the crystal structure. As expected, while a very small $\tau = 0.5$ plateaus in the case of MgO, a very large value $\tau = 5.0$ starts to get exploited for both the sample compounds, MgO and SrS. These results align with our hypothesis.

## B    DETAILS OF OUR SIMULATOR

DFT simulators, underpinned by Density Functional Theory (DFT) Parr (1989), serve as crucial computational tools for approximating the solution of the Schrödinger equation for a given system of particles. Particularly, in our study, these particles constitute the chemical structure of a crystal. By providing an approximate solution of the underlying differential equation, DFT simulators enable the calculation of system dynamics, including critical properties such as total energy, and facilitate the simulation of system relaxation to a stable, energy-minimal configuration.

DFT represents a class of computational algorithms rather than a single operation method, which justifies the availability of multiple DFT simulators. Examples of these simulators include licensed platforms like VASP, and open-source ones like GPAW Mortensen et al. (2005); Enkovaara et al. (2010). We leveraged the operational flexibility inherent to DFT in this work by using GPAW to create random stable structures as initial points for the optimization process. Its accessibility as an open-source tool, and ease of integration with Python, made it the preferred choice.

However, GPAW does have limitations, most notably the absence of pseudo-potentials for all chemical elements, essential for approximating the potential experienced by valence electrons in atoms. This limitation hindered the simulation of some structures used for evaluation, as highlighted by Cheng et al. (2022). Consequently, our evaluation was limited to 25 out of the original 29 compounds discussed in this prior work that informed our evaluation procedure.

## C EXPERIMENT DETAILS

In this section, we detail the hyperparameters and configurations employed in our experiments to facilitate reproducibility of the results. Please note that for competing models, we rely on results reported in the original work instead of replicating the experiments. For comprehensive information regarding these models, we refer the reader to the work of Cheng et al. (2022).

**Encoding and Decoding** Following the method in Xie et al. (2021), we firstly train an variational encoder to transform crystal structure to CD-VAE latent space, with the same training protocol in Xie et al. (2021). We use a batch size of 256 here when training the encoder.

**Hyperparameters** In LCOMs, we follow most of the hyper-parameters in the implementation of COMs method Trabucco et al. (2021b). The number of epochs to train the model $\widehat{E}_\theta(\phi(\mathbf{x}, c), c)$ is 50 and the number of gradient descent steps used in Equation 5 is 50. The number of steps used to generate optimized results in latent space is 10 for model trained with OQMD dataset and 40 for model trained with MatBench dataset. Please note that this number is picked by evaluating the distance between optimized groups and training dataset. The training batch size is 128 and the learning rate for model training is 0.00003. The model structure is followed by the one in Trabucco et al. (2021b). The overestimation limit $\tau$ in Equation 6 of Trabucco et al. (2021b) is picked as 1.0.

