# OpenReview forum: "Latent Conservative Objective Models for Offline Data-Driven Crystal Structure Prediction"
_ICLR.cc/2024/Conference — Submitted to ICLR 2024_

### Official Review · Reviewer_FciH · 2023-10-30

**Soundness:** 1 poor
**Presentation:** 2 fair
**Contribution:** 1 poor
**Rating:** 3
**Confidence:** 4

**Summary:**

This work proposes a method named LCOMs for predicting equilibrium 3D crystal material structure from atom compositions. The proposed LCOMs method first maps crystals to latent vectors by CDVAE models, then optimize crystal structures by a conservative surrogate model. Experiments shown that LCOMs can successfully optimize crystal structures in higher probability than traditional methods.

**Strengths:**

Originality: This work studies a relatively new and under-explored problem of crystal structure prediction.
Quality: This work conducts basic crystal structure prediction experiments on 26 crystals to show the advanges of the proposed method.
Clarity: The writing of this paper is overall clear and well-organized. Method details is well elaborated.
Significance: Promising results of crystal structure prediction on 26 crystals show that the proposed LCOMs method is somewhat valuable in crystal structure prediction.

**Weaknesses:**

Major:
(1) A major limitation of this work lies in the lack of sufficient technical novelty contribution in the aspect of machine learning method. The proposed method is largely the combination of CDVAE model [1] and COMS method [2], and the only novelty contribution is proposing to apply the optimization procedure of COMs method to the latent vectors obtained by CDVAE. Most of the methodology description is describing the two existing methods, and no description about the motivation or significance of the novelty contribution in methodology is given. Hence, this work does not reach the bar of top-tier machine learning conferences like ICLR.
(2) Though the crystal structure prediction problem is largely under-explored, there exist a method DiffCSP [3] for solving this problem. As DiffCSP also adopts generative models to encode crystal materials, authors are recommended to discuss and clarify the difference between LCOMs and DiffCSP.
(3) Experimental results are not sound enough. Experiments are only conducted on 26 materials, this test set size is too small to make reliable conclusion from experiment results. Also, in experiments, whether a crystal is successfully optimized is determined by its difference in energy to the globally optimal structure. It would be better to also report the exact difference in structure themselves, which can be measured by match rate and structure RMSE used in [1][3]. Authors are recommended to use a large test set with at least hundreds of materials and add structure difference metrics.

Minor:
The bibliography citation notations are not appropriately used in many places. Please use citation with paratheses when the citation itself is not a main part of a sentence.

[1] Crystal Diffusion Variational Autoencoder for Periodic Material Generation. ICLR 2022.
[2] Conservative Objective Models for Effective Offline Model-Based Optimization. ICML 2021.
[3] Crystal Structure Prediction by Joint Equivariant Diffusion on Lattices and Fractional Coordinates. ICLR 2023 ML4Materials Workshop.

**Questions:**

No additional questions.

---

### Official Review · Reviewer_Ffti · 2023-10-31

**Soundness:** 2 fair
**Presentation:** 2 fair
**Contribution:** 2 fair
**Rating:** 3
**Confidence:** 4

**Summary:**

In this paper, the authors propose LCOMs, applying COMs on the latent space of CDVAE [1] to address the crystal structure prediction (CSP) problem. The proposed method makes the latent space more robust for the gradient-descent optimization, leading to better performance against other optimization-based baselines.

**Strengths:**

The paper focuses on an important problem and the proposed method is simple yet effective.

**Weaknesses:**

1. The contribution is incremental. The paper directly applies the COMs technique [2] to regularize the energy prediction model from the latent space of CDVAE, without making significant modifications to either method. Although the proposed method technically achieves better results, the paper does not provide new insights or broader directions in the fields of crystal representation or latent optimization.
2. The experimental dataset is limited. The main evaluation is conducted on a small dataset of only 26 compounds, which undermines the persuasiveness of the conclusions. Additionally, some results are unclear. In Table 2, LCOMs reduces the wall-clock time to 2 seconds, as it "does not require running expensive message passing loops of a graph neural network encoder but rather runs relatively faster forward passes through small MLPs." It appears that the time calculation only considers the latent optimization phase, neglecting the decoding time, which requires a 5k-step Langevin Dynamics in the original CDVAE and should take more than 2 seconds.
3. The writing needs improvement, and some details are missing. This paper uses CDVAE to address the CSP problem. As mentioned in Section 4.1, the encoder of CDVAE is Dimenet++, which requires a 3D structure as input. However, as defined in Problem 3.1, the goal of CSP is to predict the 3D structure from the composition, which means the 3D structure is UNKNOWN. It is unclear how the authors obtain the latent representation from the encoder without 3D input. Is there any data leakage? Moreover, when introducing CDVAE, there is a mistake that CDVAE does not decode from "random lattice", but the lattice predicted from the latent state.

[1] Xie, Tian, et al. "Crystal Diffusion Variational Autoencoder for Periodic Material Generation." International Conference on Learning Representations. 2021.
[2] Trabucco, Brandon, et al. "Conservative objective models for effective offline model-based optimization." International Conference on Machine Learning. PMLR, 2021.

**Questions:**

Please see the weakness part above.

---

### Official Review · Reviewer_YfQf · 2023-10-31

**Soundness:** 3 good
**Presentation:** 3 good
**Contribution:** 2 fair
**Rating:** 6
**Confidence:** 2

**Summary:**

This paper addresses the CSP problem. The authors propose a surrogate model is trained to be conservative so as to prevent exploitation of its errors by the optimizer. The proposed method achieves the SOTA performance.

**Strengths:**

Originality: This work has some novelty in general. The idea of the surrogate model is interesting to the materials science community.

Quality: The quality of this work is above average. The technical part is sound for material science people with properly defined terms and clear problem formulation.

Clarity: This paper is clearly written. Each part of the proposed method is well-motivated.

Significance: The work solves a significant problem in material science.

**Weaknesses:**

1. More discussions on the difference between [1] and the proposed method should be added.

2. Some reference papers like [2[ should be added as well.

[1] T. Xie, X. Fu, O.-E. Ganea, R. Barzilay, and T. Jaakkola. Crystal diffusion variational autoencoder
for periodic material generation. arXiv preprint arXiv:2110.06197, 2021.

[2] Rui Jiao and Wenbing Huang and Peijia Lin and Jiaqi Han and Pin Chen and Yutong Lu and Yang Liu. Crystal Structure Prediction by Joint Equivariant Diffusion on Lattices and Fractional Coordinates. Workshop on ''Machine Learning for Materials'' ICLR 2023

**Questions:**

1. What are the main differences between [1] and the proposed method in terms of both the encoder and the decoder?

[1] T. Xie, X. Fu, O.-E. Ganea, R. Barzilay, and T. Jaakkola. Crystal diffusion variational autoencoder
for periodic material generation. arXiv preprint arXiv:2110.06197, 2021.

---

### Official Review · Reviewer_atym · 2023-11-03

**Soundness:** 3 good
**Presentation:** 3 good
**Contribution:** 1 poor
**Rating:** 3
**Confidence:** 5

**Summary:**

This paper proposes a method to optimize crystal structures called "Latent Conservative Objective Models" (LCOMs). The method is essentially conservative objective models applied in the latent space of a variational autoencoder.

I think this paper proposes a sensible method to solve a realistic problem. My main concern is that the machine learning novelty is fairly low. There is nothing wrong with the paper per se, but I think ICLR might not be the right venue for this kind of work.

**Strengths:**

- Good choice of problem
- Writing is reasonably clear
- Method proposed is a sensible combination of prior methods

**Weaknesses:**

- Novelty/significance: even though the method seems well-designed, I think it is more or less a straightforward combination of COMs and CD-VAE. Although this is probably a good choice for solving the problem, I don't think this adds new knowledge to the field of ML.
- Experiments are very small-scale: the key experiments in this paper used a dataset of only 26 test crystal structures. It is unclear if a statistically significant comparison of different algorithms can be made with such a small test set.

**Questions:**

My questions mainly follow from the weaknesses I identified:

1. I suggest ICLR  might not be the right venue for this work. Do you agree with my points? If not, what aspects of this paper do you think make it in the scope of ICLR, which is a conference about ML methods?
2. Do you have any kind of measurement of the statistical significance of the results in table 1? You say error bars are n/a, but all the algorithms studied have some randomness so I disagree with this.

---

### Meta-Review · Area_Chair_UUwC · 2023-12-06

**Metareview:**

Three of the reviewers independently called for better positioning and differentiation of the authors' work from Xie et al., and Trabucco et al., citing concerns about novelty and positioning relative to these papers. Since the authors did not submit author feedback, it's challenging to recommend the paper for acceptance or to believe that these concerns have been adequately addressed.

**Justification For Why Not Higher Score:**

The reviewers raised a couple of unanimous concerns, but the authors did not submit author feedback.

**Justification For Why Not Lower Score:**

N/A

---

### Decision · Program_Chairs · 2024-01-16

Reject